# Fast and High-Efficiency Synthesis of Capsanthin in Pepper by Transient Expression of Geminivirus

**DOI:** 10.3390/ijms241915008

**Published:** 2023-10-09

**Authors:** Zhimin Lin, Muhammad Moaaz Ali, Xiaoyan Yi, Lijuan Zhang, Shaojuan Wang

**Affiliations:** 1Fujian Academy of Agricultural Sciences Biotechnology Institute, Fuzhou 350003, China; 2College of Horticulture, Fujian Agriculture and Forestry University, Fuzhou 350002, China1210305019@fafu.edu.cn (X.Y.); 1210305020@fafu.edu.cn (L.Z.); 3210330057@fafu.edu.cn (S.W.)

**Keywords:** capsanthin, carotenoids, pepper, transient system, geminivirus

## Abstract

The color of the chili fruit is an important factor that determines the quality of the chili, as red chilies are more popular among consumers. The accumulation of capsanthin is the main cause of reddening of the chili fruit. Capsanthin is an important metabolite in carotenoid metabolism, and its production level is closely linked to the expression of the genes for capsanthin/capsorubin synthase (*CCS*) and carotenoid hydroxylase (*CrtZ*). We reported for the first time that the synthesis of capsanthin in chili was enhanced by using a geminivirus (Bean Yellow Dwarf Virus). By expressing heterologous β-carotenoid hydroxylase (*CrtZ*) and β-carotenoid ketolase (*CrtW*) using codon optimization, the transcription level of the *CCS* gene and endogenous *CrtZ* was directly increased. This leads to the accumulation of a huge amount of capsanthin in a very short period of time. Our results provide a platform for the rapid enhancement of endogenous *CCS* activity and capsanthin production using geminivirus in plants.

## 1. Introduction

Chili peppers (*Capsicum* spp.) have long captivated human senses with their fiery flavors and vibrant colors. Beyond their culinary significance, these iconic fruits have emerged as important subjects of scientific inquiry, owing to their rich repertoire of bioactive compounds [1]. The chili pepper is a very common vegetable that is rich in a variety of pigments, such as anthocyanins and carotenoids [2]. Anthocyanins are a group of polyphenolic pigments found in plants such as pepper, tomato, eggplant, and potato [3]. Carotenoids constitute a family of natural pigments spanning a spectrum from yellow to red, characterized by potent antioxidant properties. Nature has revealed over 1000 distinct carotenoid structures to date [4]. Among these compounds, capsanthin stands out as a valuable carotenoid pigment, not only for its role in determining the striking red, orange, and yellow hues of peppers but also for its potential health benefits [5,6]. Capsanthin has garnered attention for its antioxidant properties and potential health-promoting effects, making it a subject of interest in both food science and pharmaceutical research [7]. However, the efficient synthesis of capsanthin within chili peppers has remained a challenge, requiring time-consuming and resource-intensive processes [8].

Capsanthin plays a key role in the development of the distinctive red color of pepper fruit, a member of the xanthophylls, a class of oxygenated carotenoids [7]. In the biosynthetic pathway of capsanthin, the most important genes involved in the synthesis are phytoene synthase (*PSY*), lycopene β-cyclase (*LCYB*), β-carotene hydroxylase (*CrtZ*), and capsanthin/capsorubin synthase (*CCS*) [9]. Among these, capsanthin synthesis depends on the normal expression of two key enzyme genes, i.e., *CCS* and *CrtZ*. The *PSY* gene, which encodes phytoene synthase, is a key regulator of carotenoid biosynthesis in pepper [9]. The full-length *CaPSY1*, *CaPSY2*, and *CaPSY3* genes have been identified, with the expression levels of *PSY1* being directly related to the accumulation of carotenoids [10]. Two genes, *CaLCYB1* and *CaLCYB2*, have been discovered in the pepper genome, and their expression found to be associated with the presence of carotenoids [11]. The study of five cultivars, including the mutant *Oranzheva kapia*, showed that two hydroxylase genes, encoding *CrtZchr03* and *CrtZchr06*, are present in pepper on chromosomes 3 and 6, and that deletion of the *CrtZchr03* gene resulted in an increase in β-carotene accumulation [12]. Co-transformation of the genes responsible for β-carotene, 4,4′-ketolase (*CrtW*) and β-carotene 3,3′-hydroxylase (*CrtZ*), has been utilized to engineer transgenic organisms capable of producing astaxanthin, a naturally occurring carotenoid pigment known for its antioxidant properties and vibrant red-orange coloration [13]. In *E. coli*, the production of zeaxanthin was significantly enhanced by optimizing the codons of the *CrtZ*, *CrtY*, and *CrtI* genes, resulting in a 10-fold increase in zeaxanthin yield [14]. Zeaxanthin is a naturally occurring pigment known for its vibrant yellow-orange color and its potential benefits, such as its role as an antioxidant and its involvement in various physiological processes [15].

The capsanthin/capsanthin synthase (*CCS*) gene plays a pivotal role in controlling capsanthin synthesis and the development of red pigmentation in peppers [16]. The CCS enzyme is responsible for the production of the cyclopentane or κ-ring that is characteristic of the carotenoids in paprika [17]. Typically, yellow peppers have premature stop codons due to mutations in the coding region of the CSS [18]. In addition, variation in β-carotene hydroxylase2 may be related to pepper color and its elevated expression could be a regulator of carotenoid pathway genes [19]. Overexpression of beta-carotene ketogenase from *Paracoccus* sp. strains increases adenosine accumulation in *E. coli* [20]. The current production of capsanthin is concentrated on *E. coli* cells [21,22], and it has been successfully produced. The *CrtW* gene can have a direct effect on endogenous beta-carotene or zeaxanthin as precursors [23]. Previous studies have shown that the transformation of transgenic plants with the *CrtW* gene carrying the small subunit transit peptide of pea Rubisco, driven by the CAMV-35S promoter, resulted in the accumulation of new carotenoids in the leaves and petals of the plants and triggered a change in flower color [24]. Of course, in bacteria, the function of β-carotene hydroxylase (*CrtZ*) and β-carotene ketolase (*CrtW*) is mainly to convert β-carotene to astaxanthin due to differences in natural substrates [25,26,27].

In this study, we took full advantage of the high replicative expression capacity of geminivirus to demonstrate that capsanthin can be synthesized in greater quantities in less time using the plant as a factory. We have optimized the *CrtZ* and *CrtW* genes using the codon of *Lactuca sativa*, constructed them into a geminivirus backbone vector, and achieved their transient expression in pepper. The results of real-time quantitative PCR showed that the transient expression of *CrtZ* and *CrtW* genes not only increased the expression of endogenous *CrtZ0*, but also caused a significant increase in the expression level of *CCS*. However, when analyzed in terms of transient color change and *CCS* gene expression, *CrtW* expression was more effective than *CrtZ*. The results of the analysis of capsanthin and capsorubin content by HPLC-MS/MS revealed that overexpression of the *CrtW* gene in chili peppers caused a significant accumulation of capsanthin. This work, therefore, represents the first successful attempt to biosynthesize capsanthin using heterologous genes in peppers and will provide a platform for further research into the mechanism of capsanthin synthesis.

## 2. Results

### 2.1. Analysis of Geminivirus Infection in N. benthamiana Plants

We infected *Nicotiana benthamiana* plants with *Agrobacterium tumefaciens* carrying a geminivirus fusion GFP vector p1300BGFP (Appendix A). It was clearly visible that the GFP protein was strongly expressed on tobacco leaves after 3 days by UV light (Analytik Jena, Tewksbury, MA, USA). Furthermore, our observations indicated that the geminivirus infection was exclusively confined to the inoculated leaves, as there was no evidence of infection in the non-inoculated leaves, as compared to the control group (Figure 1a,b).

### 2.2. Codon Optimization and Gene Synthesis

The *CrtW* and *CrtZ* genes were both derived from *Brevundimonas* sp. strain SD212. To optimize the *CrtW* and *CrtZ* genes and increase their expression in pepper, we selected two types of plant codons from *Lactuca sativa*. We controlled the GC content of both genes at 41.33% and 40.08%, respectively. In addition, the double termination codon TAATGA had been added. Between the gene and the promoter was a signal peptide derived from ribulose bisphosphate carboxylase small chain 2A, which was named TP (Appendix A). These components were directly synthesized by the gene, including CAMV-35S and rbcs promoters. Gene synthesis was carried out by Generay Biotechnology, Shanghai, China, and preserved in plasmids.

### 2.3. Efficient Expression Vectors to Produce Capsanthin in Peppers

Two efficient geminivirus expression vectors were constructed using pepper as the host, with *CrtW* and *CrtZ* as the target genes. The *CrtW* gene was driven by the CAMV-35S promoter, while the *CrtZ* gene was driven by the rbcs promoter. By the addition of a signal peptide, the expression of the gene was increased (Figure 2A,B). With two long intergenic regions as the entire expression region, the expression of both *CrtW* and *CrtZ* genes was enhanced with continuous replication of the geminivirus, resulting in a transient increase in capsanthin production.

### 2.4. The CrtZ and CrtW Genes Speed up Color Change in Peppers

Transient synthesis of the *CrtZ* and *CrtW* genes in peppers was carried out through p1300BZ and p1300BR vectors. The results showed that *CrtZ* and *CrtW* rapidly accelerated the color change of the peppers in comparison to the control (Mock) within only 3 days (Figure 3A). The RT-qPCR results revealed that overexpression of the *CrtW* and *CrtZ* genes increased the endogenous *CCS* gene more than 11,000-fold and 3700-fold, respectively, compared to the control (Mock) (Figure 3B). Other genes such as *CrtW*, *CrtZ*, *CrtZ0*, *LCYB,* and *PSY*, which are associated with capsanthin synthesis, were differentially up-regulated in p1300BR or p1300BZ (Figure 3C,D).

### 2.5. Effect of the CrtW Gene on Capsanthin Production

The pepper powder was subjected to treatment with cell lysate and subsequently centrifuged. It was observed that both p1300BR and p1300BZ appeared darker than the control, Mock, with p1300BR displaying the most pronounced red coloration (Figure 4A). Zeaxanthin, violaxanthin, capsorubin, capsanthin and antheraxanthin were analyzed by HPLC-MS/MS. Peak absorbance was determined at 450 nm (Figure 4B). The results indicated that capsanthin production was the highest among the products, except for zeaxanthin, that exhibited a substantial increase, reaching approximately 19-fold higher levels in p1300BR when compared to the Mock sample (Figure 4C).

## 3. Discussion

Capsanthin and capsorubin are red pepper carotenoids with powerful antioxidant properties. In the case of red pepper, it contains approximately 9.15 mg of carotenoids per 100 g of fresh weight (FW), with capsanthin, the dominant pigment, making up 46% of this total [28,29]. Previously, the synthesis of other carotenoids in plants or microorganisms was mainly focused on β-carotene and astaxanthin [30]. Nevertheless, capsanthin is one of the carotenoids closely associated with human health, influencing glucose metabolism, LDL receptor expression, cholesterol catabolism and so on [31]. Previous studies have shown that antheraxanthin can be biosynthetically converted to capsanthin [32,33]. Expression of *CCS* cDNA using a viral RNA vector resulted in the accumulation of large amounts of capsanthin in *N. benthamiana* leaves [34], including the latest synthesis and production of capsanthin in *E. coli* [21]. In this study, the heterologous β-carotene hydroxylase and β-carotene ketolase were optimized mainly by plant codon optimization. We then made them work for the first time using a special genetic tool called a geminivirus fusion vector. This approach boosted the activity of the *CCS* gene, leading to a significant increase in capsanthin production.

### 3.1. Effect of Heterologous CrtZ and CrtW Genes on CCS in the Carotenoid Pathway

The *PSY*, *LCYB*, *CrtZ0*, and *CCS* genes play pivotal roles in the entire process of carotenoid synthesis, serving as key genes in this intricate biochemical pathway [35]. The *PSY* gene is a key gene upstream of the carotenoid biosynthetic pathway, and its normal expression facilitates downstream carotenoid synthesis [36]. Normal expression of endogenous CrtZ0 synthase, a key enzyme that catalyzes the formation of zeaxanthin from β-zeaxanthin, has a direct effect on capsanthin synthesis [12]. The *CCS* gene is located in the final stage of the capsanthin synthesis pathway and is an important rate-limiting enzyme gene in capsanthin formation. The two modified genes, β,β-carotenoid 3,3′-hydroxylase (*CrtZ*) and β,β-carotenoid 4,4′-ketolase (4,4′-oxygenase; *CrtW*), were derived from a marine bacterium, *Brevundimonas* sp. strain SD212. Using lettuce codon preferences, we optimized codons for two enzyme genes. Genetic studies using genes from this strain have been successful in promoting carotenoid accumulation [37], including improving biosynthesis of astaxanthin in *Escherichia coli* [38]. β-carotenoid hydroxylase (CrtZ) is the last major enzyme in the zeaxanthin biosynthetic pathway [39]. In peppers, the *CrtZchr06* gene, located on chromosome 6, converts beta-carotene into zeaxanthin [12]. Production of zeaxanthin in tobacco leaves by expression of heterologous beta-carotene hydroxylase (CrtZ) improves protection against high light and UVradiation [40]. When infested with the p1300BZ plasmid, heterologous *CrtZ* overexpression enhanced the expression of endogenous *CrtZ0* and *CCS* genes and promoted capsanthin production (Figure 3A,D). The experimental results showed that the transcriptional expression of β-carotene hydroxylase (*CrtZ*) is directly correlated with the transcriptional level of capsanthin/capsorubin synthase (*CCS*) in peppers. Currently, the involvement of *CrtW* in the zeaxanthin biosynthesis pathway is poorly understood. However, CrtW proteins can convert zeaxanthin to astaxanthin [41]. The fusion enzyme CrtZ-CrtW reduces zeaxanthin and canthaxanthin content, thereby increasing astaxanthin production [42]. In the p1300BR plasmid, the overexpression of the *CrtW* gene increased endogenous *CrtZ0* expression (Figure 3C), while the endogenous *CCS* gene was dramatically up-regulated by almost 10,000-fold or more (Figure 3B). Meanwhile, in peppers, it was shown to promote the production of capsanthin, with a more than 10-fold increase in yield compared to the control by HPLC-MS/MS (Figure 4). This is the first time we have shown that overexpression of heterologous *CrtW* increases transcript levels of *CCS* genes, thereby promoting capsanthin accumulation in pepper.

### 3.2. Realization of Chili Peppers as Reactors Using Geminiviruses

The biosynthesis of carotenoids is mainly carried out using *E. coli* [43], yeast [44], and viral vectors [45] as reactors. However, the aim of the major systems is to continuously improve the accumulation of products. Biosynthesis and accumulation of carotenoids in plants usually occurs in culture during post-harvest storage. However, such compounds are often degraded during plant senescence or processing. Therefore, it would be more valuable to use the plant fruit as a reactor to complete product accumulation during fruit development [46]. Geminivirus has a unique envelope structure, consisting of coat proteins (CPs) that form icosahedra, and it can infect a wide range of agricultural plant hosts to carry out its functions [47]. They are usually a ring of single-stranded DNA, and the genome is packaged into equally spaced twin particles. Earlier, geminivirus has been reported to synergistically infect pepper [48]. Geminivirus has the ability to efficiently express exogenous proteins, and expression of the β-glucuronidase (GUS) reporter gene using Bean Yellow Dwarf Virus (BeYDV) to construct vectors is found to result in a 40-fold increase in expression levels compared to controls [49]. In this research, we utilized the putative genes from BeYDV (GeneBank: NC_003493) to create geminivirus vectors. These vectors were designed for expressing the core or essential parts of genes. The results of fusion expression by GFP proteins show that geminiviruses differ from tobacco rattle virus in that they can only express fusion-expressed proteins at the injection site, but not in the newborn leaves (Figure 1) [50]. This type of expression allows a more targeted and efficient expression for our use of chili fruits as reactors and is confirmed by the expression of UV-irradiated GFP protein on tobacco leaves. We prepared the heterologous *CrtW* and *CrtZ* genes for transient expression in chili fruit using lettuce codon optimization. This not only helped us understand how these two enzymes control the *CCS* gene but also allowed us to make the bioreactor work effectively with chili peppers. As a result, we significantly reduced the time needed for capsanthin production. In addition, the expression effect of the co-injection of *CrtZ* and *CrtW* was poorer than that of the single injection of *CrtZ* (Appendix A). We speculate that there may be a certain competitive effect on the substrate, which needs to be further investigated in the future.

Taken together, the results reveal that the main effect of the efficient expression of heterologous *CrtW* and *CrtZ* in the upstream region of zeaxanthin is to promote the accumulation of antheraxanthin to capsanthin using the geminivirus expression system. However, they have less effect on the conversion pathway from violaxanthin to capsorubin (Figure 5). We provide a novel synthetic biology tool to rapidly achieve the regulation of key genes in metabolic pathways using a geminivirus as a vector, and to transform the target plant into a reactor for the rapid production of plant-derived compounds. Secondly, we would like to focus on promoting the use of the geminivirus system’s ability to invade between plant cells, allowing us to use key parts of the plant, such as leaves and fruits, as reactors for the rapid production of a range of important small molecule peptides or high-value antioxidants, such as astaxanthin.

## 4. Materials and Methods

### 4.1. Codon Optimization and Gene Synthesis

In order to achieve our research objective of enhancing capsanthin production in peppers, we optimized the *CrtW* and *CrtZ* genes, which are key players in the biosynthesis of this valuable carotenoid pigment. The CrtW and CrtZ were both derived from *Brevundimonas* sp. strain SD212 and contained 244 and 161 amino acids, respectively (GenBank: BDC30290.1 and QVQ68840.1). The genes had been optimized with reference to the codon preferences of *Lactuca sativa* (Appendix A). The gene sequences were synthesized by Generay Biotech, Shanghai, China and constructed on the pGH plasmid.

### 4.2. Geminivirus Expression Vector Construction

To effectively manipulate gene expression in peppers, we constructed geminivirus-based expression vectors (p1300BR, p1300BK, and p1300BZ) using the ClonExpress MultiS One Step Cloning Kit strategy (Vazyme, Nanjing, China). These vectors were strategically designed to deliver the *CrtW* and *CrtZ* genes and were derived from the Cotton leaf curl Burewala virus. The p1300BGFP was a modified insertion portion of the putative BeYDV genes V1, V2, C1, C1:C2 (Genebank:NC 003493.2) into pCAMBIA1300. The amplified product was restricted with pstI/SacI and cloned into the pstI and SacI sites of p1300BGFP, resulting in p1300BR, and p1300BZ. All of the recombinant vector constructs have been confirmed by restriction digest, polymerase chain reaction, and the associated sequence analysis.

### 4.3. Transient Expression on Tobacco and Pepper

Our research explores the impact of geminivirus-based vectors on capsanthin production in both tobacco and pepper plants. Tobacco (*Nicotiana tabacum*) and pepper (*Capsicum annuum*) were grown in an incubator at a temperature of 25 °C. Agrobacterium strain EH105 was transformed by different plasmids, i.e., p1300BGFP, p1300BR, and p1300BZ. The bacterial sediment was resuspended in NMAG solution (20 mM Na_3_PO_4_·12H_2_O, 50 mM MES, and 1 M acetosyringone, 250 μg D-Glucose), the OD600 was adjusted to 0.8, and then the resuspended bacterial solution was placed in the dark at room temperature for 2~3 h. The bacteria solution was injected with a 1 mL needle from the abaxial side of the tobacco leaf or from the cavity of the pepper.

### 4.4. Carotenoid Analysis

To assess the success of capsanthin production, we performed detailed carotenoid analysis. This analysis not only quantified capsanthin but also evaluated other carotenoids in the treated peppers. Carotenoid fractions were determined using an Agilent 1290 HPLC coupled to an AB Qtrap 6500 mass spectrometer. Specific carotenoid compounds were separated on a reversed-phase CORTECS UPLC C18+ carotenoid column (1.6 μm, 2.1 × 75 mm^2^) using a linear gradient with a mobile phase consisting of 0.1% formic acid in water (solvent A) and methanol (solvent B). The gradient elution was from 0 to 3 min for 10% A and 90% B, followed by 0% A and 100% B for 10–14 min, and 10% A and 90% B for 14.1–16 min. The flow rate was 0.45 mL/min and the column temperature was maintained at 35 °C. The elution peaks were monitored at 450 nm. The contents of carotenoid types, i.e., zeaxanthin, β-cryptoxanthin, lycopene, α-carotene, phytoene, violaxanthin, capsorubin, capsanthin, astaxanthin, antheraxanthin were calculated from their corresponding calibration curves (zeaxanthin (y = 10.15358x + −200.30749; r = 0.99353), β-cryptoxanthin (y = 24.89303x + −115.05656; r = 0.99507), lycopene (y = 74.23651x + −264.74526; r = 0.99413), α-carotene (y = 50.67507x + 58.01636; r = 0.99822), phytoene (y = 1426.27367x + 318.75576; r = 0.99853), violaxanthin (y = 65.01299x + −5283.14504; r = 0.99785), capsorubin (y = 1606.31637x + −6.61120 × 10^4^; r = 0.99354), capsanthin (y = 189.37391x + −868.45427; r = 0.99172), astaxanthin (y = 376.66x − 575.31; r = 0.9955); antheraxanthin (y = 69.439x − 6652.2; r = 0.998)) (Appendix A).

### 4.5. RNA Extraction and RT-qPCR Analysis

To delve deeper into the molecular mechanisms behind capsanthin production, we conducted RNA extraction and RT-qPCR analysis on pepper fruits. Total RNA extraction of pepper fruits was performed according to the instructions of the Polyphenol Total RNA Kit (TIANGEN, Beijing, China, No. DP441), and the cDNA was synthesized using a reverse transcriptase kit (Vazyme, Nanjing, China, No. R323) for RT-qPCR. The ubiquitin gene was used as the reference gene. RT-qPCR analysis was performed using a QuantStudio 1 (ABI) machine and AceQ Universal SYBR qPCR Master Mix Kit (Vazyme, Nanjing, China) to determine the relative expression levels of target genes. Gene-specific primers were designed based on the coding region sequence of each gene using Primer 6.0 software. Three replicates were run for each sample. We quantified the relative changes in gene transcript levels using the 2^−ΔΔCT^ method [51]. Appendix A lists all primers used in this study.

## 5. Conclusions

In conclusion, our study has yielded significant insights into the utilization of geminivirus-based vectors to optimize the expression of the *CrtW* and *CrtZ* genes in peppers, ultimately leading to a substantial increase in capsanthin production. The comprehensive approach included codon optimization, careful control of GC content, and the incorporation of signal peptides, all of which contributed to the remarkable enhancement of gene expression. By utilizing two distinct promoters, CAMV-35S and rbcs, in the constructed expression vectors, we were able to create an efficient system for transiently elevating capsanthin levels in peppers. This resulted in a rapid color change in the treated peppers, significantly faster than was observed in the control group. Furthermore, our research revealed a substantial increase in zeaxanthin levels, underscoring the broader impact of this approach on the overall nutritional composition of peppers. These findings hold great promise for both the agricultural and food industries, as they offer a novel avenue for increasing the yield of valuable compounds in crops and enhancing their nutritional value. Additionally, the specific and localized geminivirus infection in *N. benthamiana* plants underscores the safety and precision of this method, providing a strong foundation for future research in crop enhancement and the development of nutritionally enriched foods.

## Figures and Tables

**Figure 1 ijms-24-15008-f001:**
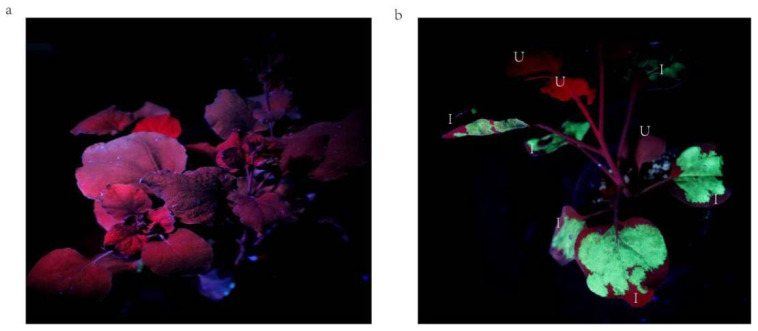
Visualization of GFP expression in *N. benthamiana* plants under UV light. (**a**). Normal tobacco plants under UV light; (**b**). GFP protein expression on tobacco leaves 3 days after injection under UV light. I, inoculated leaves; U, uninoculated leaves.

**Figure 2 ijms-24-15008-f002:**
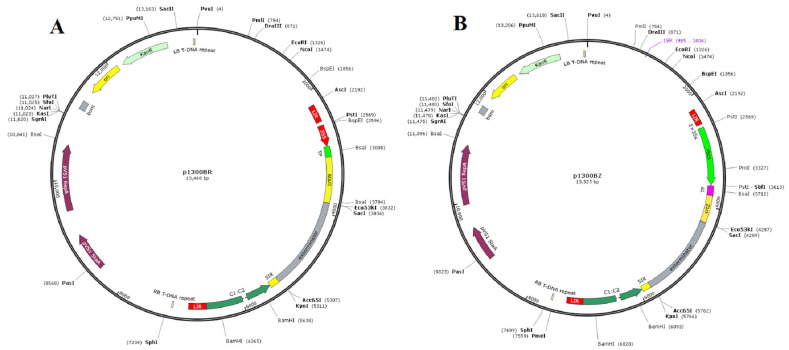
The *CrtW* and *CrtZ* genes are expressed in a geminivirus expression system under the CAMV-35S promoter and the strong rbcs promoter. (**A**) The p1300BR vector was used for *CrtW* gene expression; (**B**) the p1300BZ vector was used for *CrtZ* gene expression.

**Figure 3 ijms-24-15008-f003:**
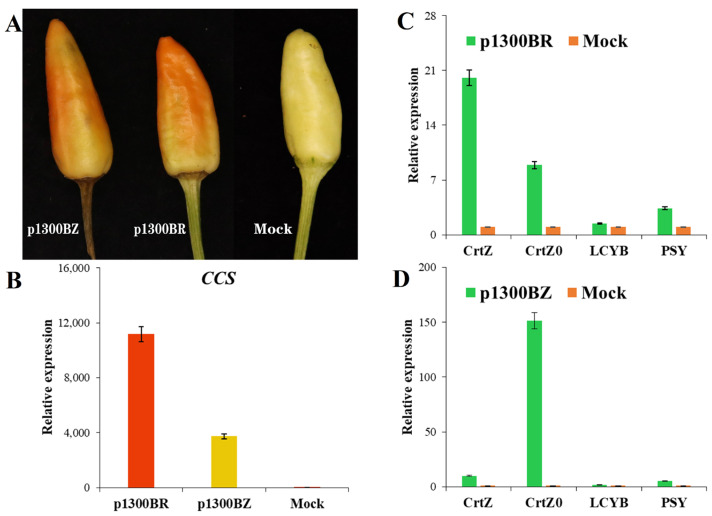
The *CrtZ* and *CrtW* genes promote the ripening of the pepper while increasing the expression of the *CCS* gene. (**A**) The phenotypic observation of p1300BZ and P1300BR, with Mock as a control; (**B**) RT-qPCR expression analysis of the *CCS* genes in p1300BR, p1300BZ, and Mock; (**C**) RT-qPCR analysis of the expression of the genes *CrtW*, *CrtZ0*, *LCYB*, *PSY* in p1300BR and Mock; (**D**) RT-qPCR analysis of the expression of the genes *CrtZ*, *CrtZ0*, *LCYB*, *PSY* in p1300BZ and Mock. All gene expression analyses were performed using ubiquitin as an internal reference.

**Figure 4 ijms-24-15008-f004:**
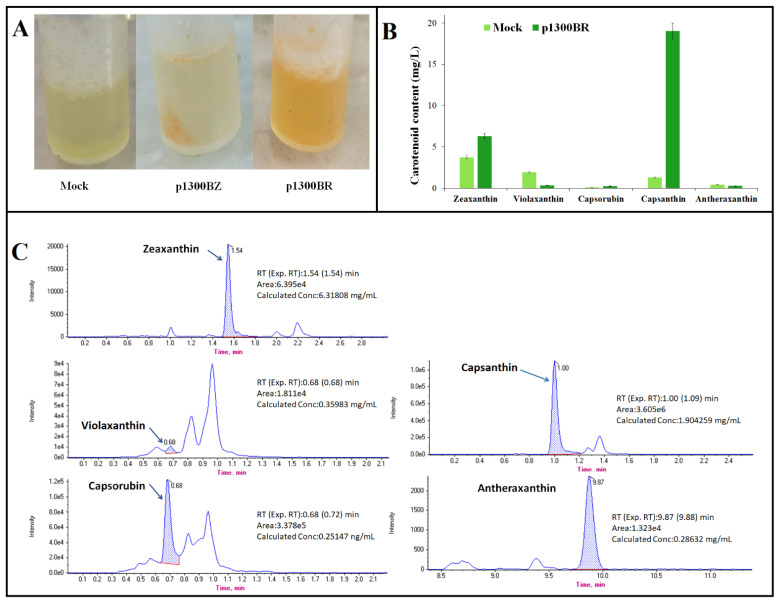
Expression of *CrtW* gene significantly enhances capsanthin production in pepper in a geminivirus system. (**A**) Color comparison of three different pepper powders in cell lysate, including Mock, p1300BZ, and p1300BR; (**B**) carotenoid production of zeaxanthin, violaxanthin, capsorubin, capsanthin, and antheraxanthin in Mock and p1300BR; (**C**) values were calculated from the peak area of the HPLC chromatogram.

**Figure 5 ijms-24-15008-f005:**
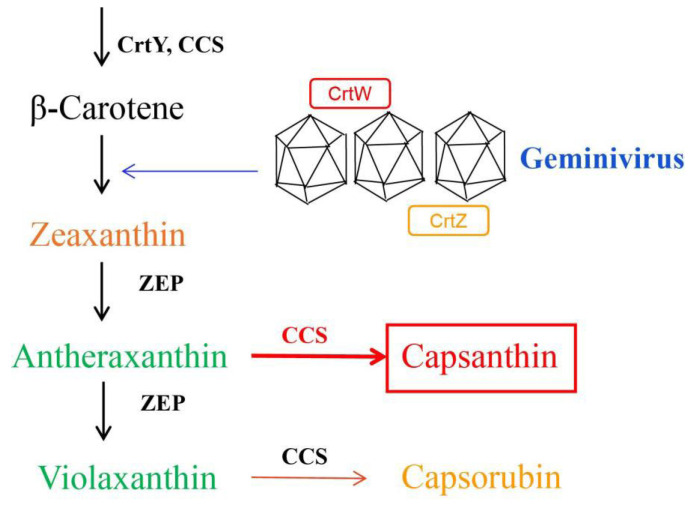
Synthesis pattern of capsanthin under a geminivirus expression system. CrtW—β-carotenoid ketolase; CrtY—lycopene β-cyclase; CrtZ—β-carotenoid hydroxylase; CCS—capsanthin/capsorubin synthase; ZEP—zeaxanthin epoxidase.

## Data Availability

Not applicable.

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
