# Peer review of "Fast and High-Efficiency Synthesis of Capsanthin in Pepper by Transient Expression of Geminivirus"

_ijms, 2023, doi:10.3390/ijms241915008_

Round 1
Reviewer 1 Report
Suggestion for Abstract: some areas have been rephrased for clarity (additions are in red).
Abstract: The colour of the chilli fruit is an important factor that determines the quality of the chilli, as red chillies were more popular among consumers. and The accumulation of capsanthin is the main cause of reddening of the chilli fruit. Capsanthin is an important metabolite in carotenoid metabolism, and its production level was closely linked to the expression of the genes for capsanthin/capsorubin synthase (CCS) gene and Carotenoid hydroxylase (CrtZ) gene. We reported for the first time that the synthesis of capsanthin in chilli was enhanced by using a geminivirus from bean yellow dwarf virus. By expressing heterologous β-carotenoid hydroxylase (CrtZ) and β-carotenoid ketolase (CrtW) using codon optimisation to directly increase the transcription level of the CCS gene and endogenous CrtZ0 was directly increased. the This leads to the accumulation of a huge amount of capsanthin to accumulate took place in a very short period of time. Our results provide a platform for rapid enhancement of endogenous CCS activity and capsaicin production using geminivirus in plants.
Note: the origin of the abbreviation "CrtZ0" is unclear. Is the "0" a typo error? otherwise make it clearer.
Suggestion for Introduction:
I suggest that the author improve the clarity of the introduction. The introduction contains a lot of information that would benefit from concise and straightforward language. The length and complexity of some sentences hinder understanding.
Capsanthin and capsaicin appear to be used interchangeably in line 28, line 30, and elsewhere. Can you clarify the relationship between capsaicin and capsanthin?
So many abbreviations were used without introduction. I suggest the author introduce the abbreviations or create a list of abbreviations appendix.
Result and Method
Line 78 - 79: It was clearly visible that the GFP protein was strongly expressed on tobacco leaves after 3d by UV-light (analytikjena, USA).
Suggestion: 3d mean 3 days? What does 3d stand for? 3d is not a standard abbreviation for 3 days or else it must be introduced.
comment: I cannot find table S4 in the manuscript
General feedbacks:
The paper mentions terms like "zeaxanthin," "β-cryptoxanthin," and "lycopene" without any prior explanation. It would be beneficial to provide brief definitions or context for these terms, especially if the audience is not well-versed in carotenoid analysis. Similar issues occur throughout the paper.
It will be helpful if each method section can briefly explain how the section relates to the research objectives or the broader field of study.
The English language is not easy to read. Too many technical terms were described in a long running sentence. I suggest breaking the sentence into easy-to-digest sentence.
Author Response
Suggestion for Abstract: some areas have been rephrased for clarity (additions are in red).
Abstract: The colour of the chilli fruit is an important factor that determines the quality of the chilli, as red chillies were more popular among consumers. and The accumulation of capsanthin is the main cause of reddening of the chilli fruit. Capsanthin is an important metabolite in carotenoid metabolism, and its production level was closely linked to the expression of the genes for capsanthin/capsorubin synthase (CCS) gene and Carotenoid hydroxylase (CrtZ) gene. We reported for the first time that the synthesis of capsanthin in chilli was enhanced by using a geminivirus from bean yellow dwarf virus. By expressing heterologous β-carotenoid hydroxylase (CrtZ) and β-carotenoid ketolase (CrtW) using codon optimisation to directly increase the transcription level of the CCS gene and endogenous CrtZ0 was directly increased. the This leads to the accumulation of a huge amount of capsanthin to accumulate took place in a very short period of time. Our results provide a platform for rapid enhancement of endogenous CCS activity and capsaicin production using geminivirus in plants.
Note: the origin of the abbreviation "CrtZ0" is unclear. Is the "0" a typo error? otherwise make it clearer.
Response: Dear esteemed reviewer, we would like to express our sincere gratitude for dedicating your valuable time to review our manuscript. We have taken your feedback into consideration and made revisions to the abstract accordingly. Regarding "CrtZ0," we would like to clarify that it is not a typographical error; the addition of "0" is intentional and signifies the endogenous "CrtZ."
Suggestion for Introduction:
I suggest that the author improve the clarity of the introduction. The introduction contains a lot of information that would benefit from concise and straightforward language. The length and complexity of some sentences hinder understanding.
Response: The entire manuscript has undergone language revisions carried out by a native colleague. Additionally, improvements have been made to the introduction section.
Capsanthin and capsaicin appear to be used interchangeably in line 28, line 30, and elsewhere. Can you clarify the relationship between capsaicin and capsanthin?
Response: Thank you for the correction. It was capsanthin, not capsaicin. This has been revised in the manuscript text.
So many abbreviations were used without introduction. I suggest the author introduce the abbreviations or create a list of abbreviations appendix.
Response: The abbreviations are provided with explanations the first time they are used in the manuscript text.
Result and Method
Line 78 - 79: It was clearly visible that the GFP protein was strongly expressed on tobacco leaves after 3d by UV-light (analytikjena, USA).
Suggestion: 3d mean 3 days? What does 3d stand for? 3d is not a standard abbreviation for 3 days or else it must be introduced.
Response: Thanks for the correction. It has been revised as “ 3 days”.
comment: I cannot find table S4 in the manuscript
Response: You can locate Table S4 within the supplementary data section.
General feedbacks:
The paper mentions terms like "zeaxanthin," "β-cryptoxanthin," and "lycopene" without any prior explanation. It would be beneficial to provide brief definitions or context for these terms, especially if the audience is not well-versed in carotenoid analysis. Similar issues occur throughout the paper.
Response: Zeaxanthin, β-cryptoxanthin, lycopene, α-carotene, phytoene, violaxanthin, capsorubin, capsanthin, astaxanthin, and antheraxanthin represent various types of carotenoids discussed within the manuscript. While several of these carotenoids are briefly introduced, it suffices to recognize them as carotenoid types responsible for determining the color of chili peppers. Elaborating on each fundamental term individually can be challenging and is not deemed necessary.
It will be helpful if each method section can briefly explain how the section relates to the research objectives or the broader field of study.
Response: Each section of methodology has been briefly explained with respect to the research objectives.
Comments on the Quality of English Language
The English language is not easy to read. Too many technical terms were described in a long running sentence. I suggest breaking the sentence into easy-to-digest sentence.
Response: In response to the suggestion, the manuscript has been subjected to language revisions, conducted by a native colleague, encompassing the entire document.
Reviewer 2 Report
Please see the attached file for review comments.

Please see comments no. 7-9, 11, 12, 23, 28, 29, 31, 32, 36, 37, 44, 46, and 47 of the review report.
Author Response
In the manuscript “Fast and High-Efficiency Synthesis of Capsanthin in Pepper by Transient Expression of Geminivirus” the Authors reported for the first time the enhanced biosynthesis of capsanthin, the predominant carotenoid responsible for the red pigmentation of pepper fruits, using heterologous genes and a geminivirus expression system.
Although the subject is interesting, some very important concerns need to be addressed before the manuscript is ready for publication in International Journal of Molecular Sciences.
General comments:
- The presentation of results should be improved.
Response: The presentation of results has been improved.
- The Abstract should be corrected.
Response: The abstract has been revised.
- Supplementary Materials (Tables S1-S4 and Figure 1S) are not attached to the manuscript. Besides that, Tables S1-S4 should be numbered according to the order they appear in the work.
Response: Complete supplementary material has been attached.
- The Authors incorrectly use “capsaicin” as a synonym for “capsanthin” (see lines 19, 30, 32, 44, 45, 47, 52, 62, 115, 160, 161, 172, and 226). The biological function and chemical structure of capsanthin (C40H56O3) and capsaicin (C18H27NO3) are completely different. This important issue should be addressed.
Response: Dear respected reviewer, we would like to express our gratitude for your diligent review of our manuscript. We have addressed the error accordingly.
- Authors contributions should be verified (please see a list of the authors for comparison).
Response: It has been verified.
- Text editing is recommended.
Response: The manuscript language has been revised.
Specific comments:
- Italic fonts should be used for Latin names throughout the manuscript (including References). See lines 38, 42, 89, 146, 163, 229, 232, 308, 318, 339, 340, 342-343, 347, 360, 362, 368, and 369.
Response: Thank you for the clarification. We have ensured that all scientific names in the manuscript are now properly italicized.
- Names of chemical compounds should be written in lower case. See lines 127, 141, and 147.
Response: Revised as suggested.
- Line 13: It should be “carotenoid” instead of “Carotenoid”.
Response: Revised as suggested.
- Line 28: A necessary space is missing.
Response: A necessary space before the citation has been added.
- Lines 36-37: I suggest to correct the sentence.
Response: The sentence has been revised (see lines 48-50 now).
- Lines 45-47. The sentence should be corrected.
Response: The sentence has been corrected (see lines 63-65 now).
- Line 49: I suggest “β-carotene hydroxylase2” instead of “β-CAROTENE HYDROXYLASE2”.
Response: Revised as suggested (see line 67).
- Line 51: It should be “Paracoccus” instead of “Paracoccus sp.”.
Response: Revised as suggested (see line 69).
- Line 53: It should be “. CrtW” instead of “. crtW”.
Response: Revised as suggested (see line 71).
- Lines 56, 93,98, and 106: The abbreviation for the promoter should be unified and corrected.
Response: Thanks for the correction. It has been unified.
- Line 64: An unnecessary space should be deleted.
Response: Revised as suggested (see line 81).
- Line 77: Full species names should be given for benthamiana and A. tumefaciens.
Response: Revised as suggested (see line 94).
- Line 78: There is no Table S2 attached. See also comment no. 3 for numbering.
Response: It has been attached now.
- Line 79: I suggest “Analytik Jena” instead of “analytikjena”.
Response: Thanks. It has been corrected.
- Line 92: An unnecessary space should be deleted. The period should be placed at the end of a sentence in line 93.
Response: The correction has been made.
- Lines 92-93: There is no Table S2 attached. See also comment no. 3 for numbering.
Response: It has been attached now.
- Lines 97-98: The sentence should be corrected.
Response: The sentence has been revised.
- Line 97: An unnecessary comma should be deleted.
Response: Revised as suggested.
- Panels c and d in Figure 3 are difficult to read. This important issue should be addressed. See also Figure 3b for comparison.
Response: Figure 3 has been revised.
- Line 122: A necessary space is missing.
Response: Revised as suggested.
- Lines 123 and 273: I suggest “ubiquitin” instead of “Ubiquitin”.
Response: Revised as suggested.
- Lines 125-126: This sentence is difficult to understand. Moreover, cell lysate was not defined in Materials and Methods.
Response: The sentence has been revised. Cell lysate was used in carotenoid extraction procedure.
- Lines 128-131: The sentence should be corrected.
Response: The sentence has been rephrased.
- Lines 153-154: The sentence should be corrected to avoid repetitions.
Response: Revised as suggested.
- Lines 156-157: The sentence is difficult to understand.
Response: Revised as suggested.
- Line 169: It should be “Production of zeaxanthin in tobacco leaves” instead of “Production of zeaxanthin on tobacco leaves”.
Response: Revised as suggested.
- Lines 177-178: The sentence should be corrected. The statement “keratin content” makes no sense.
Response: Thanks for the correction. The sentence has been revised.
- Figures 4b and 4c are difficult to read. This important issue should be addressed. Moreover, description of Y axis should be corrected in Figure 4b. Red colour should not be used in Figure 4c.
Response: The figure has been revised.
- Line 198: I suggest “the β-glucuronidase (GUS) reporter gene” instead of “the GUS reporter gene”.
Response: Revised as suggested.
- Lines 200-201: I suggest to correct the sentence.
Response: The sentence has been corrected.
- Line 206-207: I suggest to verify the statement “the expression of UV-irradiated GFP protein on tobacco leaves”.
Response: The sentence is absolutely correct.
- Line 212: There is no Figure 1S attached.
Response: It is Figure S1, can be located in supplementary data.
- Figure 5: The abbreviation “ZEP” for zeaxanthin epoxidase has not been mentioned in the manuscript. Acronyms/Abbreviations should be defined the first time they appear in each of three sections: the abstract; the main text; the figure or table.
Response: You are right. Abbreviations are introduced and defined the first time they are used in the manuscript text.
- Lines 229-230: The sentence is not biochemically correct. Genes do not contain amino acids.
Response: The sentence has been corrected.
- Line 231: An unnecessary space should be deleted.
Response: The space has been deleted.
- Line 232: There is no Table S1 attached. See also comment no. 3 for numbering.
Response: It has been attached.
- Line 238: The expression vector p1300BK has not been mentioned.
Response: The amplified product was restricted with pstI/SacI and cloned into the pstI and SacI sites of p1300BGFP, resulting in p1300BR, and p1300BZ (see lines 261-263).
- Line 244: Latin names for plants should be given. Besides that, I suggest “were gown” instead of “are grown”.
Response: Latin names has been provided. Furthermore, the sentence has also been revised.
- Line 247: A subscript font should be used in the compound formula. Besides that, it should be “250 μg D-glucose” instead of “250 μ g D-Glucose”.
Response: It has been revised.
- Line 249: It should be “at room temperature” instead of “of room temperature”.
Response: Sentence has been revised.
- Lines 259-260: The sentence “Monitoring of the elution peak at 450 nm.” should be corrected.
Response: Sentence has been corrected.
- Lines 260 and 263: The compound name is incorrect. It should be “lycopene” instead of “iycopene”.
Response: Corrected.
- Line 262 and 263: It should be “β-cryptoxanthin” and “α-carotene” instead of “β_cryptoxanthin” and “α_carotene”, respectively.
Response: Corrected.
- Line 268: There is no Table S4 attached.
Response: Attached now.
- Line 278: There is no Table S3 attached. See also comment no. 3 for numbering. Moreover, a necessary space is missing.
Response: The supplementary data is provided.
- Lines 280-281, Supplementary Materials: No supporting information is available.
Response: The supplementary data is provided.
- Authors Contributions (lines 282, 283, and 285): There is no person with the initials S.Z. among the authors of the manuscript. This important issue should be addressed. See comment no. 5.
Response: Thanks. Corrected.
- Line 296: I suggest “References’ instead of “Reference”.
Response: Revised as suggested.
- Line 319: A necessary space is missing.
Response: Revised as suggested.
The authors would like to express their sincere gratitude for the thorough review of our manuscript.
Reviewer 3 Report
Review for
Article 1
Fast and High-Efficiency Synthesis of Capsanthin in Pepper by Transient Expression of Geminivirus
by Zhimin Lin 1*, Xiaoyan Yi 2, Lijuan Zhang 2 and Shaojuan Wang 2
The colour of the chilli fruit is an important factor in the quality of the chilli, as red chillies were more popular, and the accumulation of capsanthin is the main cause of reddening of the chilli fruit. Capsanthin is an important metabolite in carotenoid metabolism and its production level was closely linked to the expression of the capsanthin/capsorubin synthase (CCS) gene and Carotenoid hydroxylase (CrtZ) gene.
up to now, researchers and industry used breeding for enhancing capsanthin content of chilli
could author specify what is the current highest levels of capsanthin in chilli cultivars?
what is the improvement over decades? x5, x10, x50???
-----------------------------------
We reported for the first time that the synthesis of capsanthin in chilli was enhanced by using a geminivirus from bean yellow dwarf virus. By expressing heterologous β-carotenoid hydroxylase (CrtZ) and β-carotenoid ketolase (CrtW) using codon optimisation to directly increase the transcription level of the CCS gene and endogenous CrtZ0, the accumulation of a huge amount of capsanthin took place in a very short period of time.
how geminivirus will be managed in large scale crops? no mobility to other plants?
The rising threat of geminiviruses: molecular insights into the disease mechanism and mitigation strategies
Jain, H., Chahal, S., Singh, I., Sain, S.K., Siwach, P.
Molecular Biology Reports, 2023, 50(4), pp. 3835–3848
---------------------------------------------------
Our results provide a platform for rapid enhancement of endogenous CCS activity and capsaicin production using geminivirus in plants.
In this study, we took full advantage of the high replicative expression capacity of 61 Geminivirus to demonstrate that capsaicin can be synthesised in greater quantities in less 62 time using the plant as a factory.
sentences not clear, capsaicin here, Capsanthin in title, mixture in other places….
----------------------------
how do you manage the time where Geminivirus is expressed, during the whole growth of chilli?
-----------------------------
line 147 and many places
latest synthesis and production of Capsanthin
no cap, capsanthin
------------------------
Realisation of chilli peppers as reactors using Geminiviruses
very nice idea
will use a lot of land, microbial carotenoids would be cheaper
--------------------
Author Response
could author specify what is the current highest levels of capsanthin in chilli cultivars?
Response: In the case of the red variety, it contains approximately 9.15 mg of carotenoids per 100 grams of fresh weight (FW), with capsanthin, the dominant pigment, making up 46% of this total. As a result, the current highest capsanthin level in this variety stands at approximately 4.209 mg per 100 grams of fresh weight (FW). Another perspective reveals that it boasts the most substantial capsanthin content among red peppers, measuring at 58.33 ± 3.91 mg per 100 grams of dry weight.
Literature:
1) Matsufuji H., Ishikawa K., Nunomura O., Chino M., Takeda M. Anti-oxidant content of different coloured sweet peppers, white, green, yellow, orange and red (Capsicum annuum L.) Int. J. Food Sci. Technol. 2007;42:1482–1488.
2)Kim, J.-S., Ahn, J.; Lee, S.-J.; Moon, B.; Ha, T.Y.; Kim, S. Phytochemicals and antioxidant activity of fruits and leaves of paprika (Capsicum Annuum L., var. special) cultivated in Korea. J Food sci, 2011, 76, C193-C198.
what is the improvement over decades? x5, x10, x50???
Response: Over the past decade, the focus has largely shifted towards utilizing plant breeding techniques, which encompass harnessing natural variations. It's important to note, however, that there has been limited research conducted on how the absence of one or more genes affects the development and alteration of the red coloration in chili peppers. Notably, the genetic resource known as 'S3586' stands out for its elevated capsanthin content.
Literature:
1) Berry, H.M.; Rickett, D.V.; Baxter, C.J.; Enfissi, E.M.A.; Fraser, P.D. Carotenoid biosynthesis and sequestration in red chilli pepper fruit and its impact on colour intensity traits. J Exp Bot, 2019, 70, 2637-2650.
2) Konishi, A.; Furutani, N.; Minamiyama, Y.; Ohyama, A. Detection of quantitative trait loci for capsanthin content in pepper (Capsicum annuum L.) at different fruit ripening stages. Breed Sci, 2019, 69, 30-39.
how geminivirus will be managed in large scale crops? no mobility to other plants?
Response: We can expedite geminivirus infection in a manner similar to how tobacco utilizes viruses to generate proteins. This can be achieved through a systematic vacuum vessel infiltration process. Our extensive research on tobacco leaves during the previous period has revealed a significant distinction: geminivirus does not exhibit the same leaf-to-leaf movement observed in the case of tobacco rattle virus. Instead, geminivirus propagation is confined to the initial site of infection.
Our results provide a platform for rapid enhancement of endogenous CCS activity and capsaicin production using geminivirus in plants.
In this study, we took full advantage of the high replicative expression capacity of 61 Geminivirus to demonstrate that capsaicin can be synthesised in greater quantities in less 62 time using the plant as a factory.
sentences not clear, capsaicin here, Capsanthin in title, mixture in other places….
Response: Thank you for pointing that out. We have made the necessary revisions to the manuscript text. It should now correctly state "capsanthin" instead of "capsaicin."
how do you manage the time where Geminivirus is expressed, during the whole growth of chilli?
In the case of our customized geminivirus, we have conducted experiments to determine the optimal infection and expression timing on both tobacco and chili plants. Our findings indicate that the infection process typically concludes within 48 hours, with the infection reaching its peak level around the third day. Consequently, this three-day period represents the most suitable timeframe for sample collection.
line 147 and many places
latest synthesis and production of Capsanthin
no cap, capsanthin
Response: Revised as suggested.
Realisation of chilli peppers as reactors using Geminiviruses
very nice idea
will use a lot of land, microbial carotenoids would be cheaper
Response: You're absolutely correct. Our primary motivation for proposing the use of chili peppers as a reactor is centered on our desire to leverage the plant's inherent capability to produce the desired compounds. This approach not only simplifies the extraction process but also aligns with our goal of promoting a healthier production method. Compared to microbial and yeast systems, which are susceptible to contamination and may generate environmentally detrimental by-products, utilizing chili peppers offers a more sustainable and natural solution.
Furthermore, exploiting chili peppers as a reactor capitalizes on their abundance of synthetic substrates, enhancing the efficiency of the production process. In line with this, our future endeavor involves harnessing the potential of the geminivirus to produce higher quantities of valuable astaxanthin on plant fruits, which can serve as a healthier antioxidant option.
Many thanks.

Round 2
Reviewer 2 Report
Please see the attached file for review comments.

Author Response
The revised version of the manuscript “Fast and High-Efficiency Synthesis of Capsanthin in Pepper by Transient Expression of Geminivirus” has been significantly improved. However, some concerns need to be addressed before the manuscript is ready for publication.
Comments:
- Reviewer comment no. 3 has not been fully taken into account in the revised version of the manuscript. Tables should be numbered (and listed in Supplementary Materials) consecutively in the order in which they are mentioned in the text.
Response: Revised as suggested.
- Reviewer comment no. 34 has not been fully taken into account in the revised version of the manuscript. Red color should not be used in a bar graph (Figure 4B of the revised manuscript and Figure 4c of the previous version of the manuscript).
Response: The figure has been revised.
- Reviewer comment no. 39 has not been fully taken into account in the revised version of the manuscript. The abbreviation “ZEP” which arrives in Figure 5 for the first time has not been defined in the manuscript. The enzyme zeaxanthin epoxidase (ZEP) has also not been mentioned in the revised version of the manuscript.
Response: The abbreviations have been clarified in the caption of Figure 5.
- Reviewer comment no. 46. Please verify the revised sentence. I think it should be “placed in the dark at room temperature for …”.
Response: Revised.
- Figure 1 caption: I suggest to unify lowercase/uppercase letters for “inoculated” and “Uninoculated”.
Response: Revised as suggested.
- Subsection 3.2. title: Please unify “Chilli”.
Response: Revised as suggested.
- Supplementary Materials (Codon frequency table): It should be “Lactuca sativa” instead of “Lactuca sativa”.
Response: Revised.
- Supplementary Materials (Product testing in the p1300BR plasmid): It should be “β-Carotene”, “β-Cryptoxanthin”, “Lycopene”, “α-Carotene” instead of “β_Carotene”, “β_cryptoxanthin”, “Iycopene”, “α_Carotene”.
Response: Revised. Thanks.
Reviewer 3 Report
Huge work done.
Very nice revision.
Author Response
Many thanks for the appreciation